# Crystallization in Zirconia Film Nano-Layered with Silica

**DOI:** 10.3390/nano11123444

**Published:** 2021-12-19

**Authors:** Brecken Larsen, Christopher Ausbeck, Timothy F. Bennet, Gilberto DeSalvo, Riccardo DeSalvo, Tugdual LeBohec, Seth Linker, Marina Mondin, Joshua Neilson

**Affiliations:** 1Department of Physics and Astronomy, University of Utah, 115 South 1400 East, Salt Lake City, UT 84112, USA; christopherausbeck@gmail.com (C.A.); riccardo.desalvo@gmail.com (R.D.); tugdual.lebohec@gmail.com (T.L.); 2Department of Physics and Astronomy, California State University, 5151 State University Drive, Los Angeles, CA 90032, USA; tfbennet@usc.edu (T.F.B.); sethdlinker@gmail.com (S.L.); Marina.Mondin@calstatela.edu (M.M.); 3RicLab, 1650 Casa Grande Street, Pasadena, CA 91104, USA; gil@ric-lab.com; 4Department of Engineering, University of Sannio at Benevento, C.so Garibaldi 107, Pal. dell’Aquila Bosco-Lucarelli, and INFN, Sezione di Napoli Gruppo Collegato di Salerno, Piazza Guerrazzi, 82100 Benevento, Italy; joshua.neilson@gmail.com; 5Gran Sasso Science Institute, Viale Francesco Crispi, 7, 67100 L’Aquila, Italy

**Keywords:** thin film, nano-layering, coating, noise, annealing, XRD

## Abstract

Gravitational waves are detected using resonant optical cavity interferometers. The mirror coatings’ inherent thermal noise and photon scattering limit sensitivity. Crystals within the reflective coating may be responsible for either or both noise sources. In this study, we explored crystallization reduction in zirconia through nano-layering with silica. We used X-ray diffraction (XRD) to monitor crystal growth between successive annealing cycles. We observed crystal formation at higher temperatures in thinner zirconia layers, indicating that silica is a successful inhibitor of crystal growth. However, the thinnest barriers break down at high temperatures, thus allowing crystal growth beyond each nano-layer. In addition, in samples with thicker zirconia layers, we observe that crystallization saturates with a significant portion of amorphous material remaining.

## 1. Introduction

The Laser Interferometer Gravitational-Wave Observatory (LIGO) [1], Virgo [2], Kamioka Gravitational-Wave Detector (KAGRA) [3] and GEO600 [4] gravitational wave detectors perform some of the most precise interferometric measurements across a wide band of frequencies [5]. The quantum statistics of light [6] and the reflective optical coatings of the test masses [7,8] primarily set the noise budget in the mid-range frequency of these detectors. The optical coatings contribute noise in two major ways:

i. Millions of sub-wavelength-sized objects within dielectric coating layers scatter a few parts per million of the interferometer’s laser light power. We can observe these objects through the off-axis imagining of the end test mass coating surfaces illuminated by the Fabry Perot stored beams [9]. Substrate figure defects and coating defects contribute to the remaining scatter. The light scattered out of the interferometer reduces the circulating power and degrades the squeezed vacuum performance [10]. If this light was simply lost, it would amount to a relatively modest noise contribution. However, a small fraction of this scattered light is diffused back from the vacuum tank and onto the mirrors, and from there it is re-scattered into the beam with a random phase, thus injecting a significant phase noise [11,12]. Sufficiently effective optical baffles, which are difficult to install, can moderate this secondary re-scattering noise contribution [13,14,15]. However, it would be preferable to eliminate primary scattering in the first place. Even a modest reduction in scattering could be significant as it can enter twice in the phase noise generation process.

ii. The Brownian motion of the coating surface results in thermal noise, in proportion to the mechanical loss. This noise can be reduced through a number of strategies [16,17]. The dielectric mirror coatings on the current test masses are made from alternating layers of silica and titania-doped tantala, two materials with, respectively, low and high indices of refraction. Silica and titania-doped tantala were selected to reduce mechanical loss, however, this can be improved [18]. The high refraction index material, which is more prone to crystallization, was observed to contribute most of the mechanical loss responsible for the thermal noise [19]. To improve reflectivity and suppress thermal noise, the pair of materials chosen, their respective thicknesses, and the number of doublet layers have thus far been collectively optimized [20,21]. The development of high-refraction index materials with lower mechanical loss is needed to further reduce the thermal noise. Progress towards achieving this goal is being made through various approaches such as experimentation with doping, different deposition and annealing methods, and novel high-index material candidates [22,23,24]. One such material is Zirconia, which has been shown to have potential as a promising replacement for current coatings [25]. Nano-layering is a technique to produce structured materials that are observed to crystallize at higher temperature and can be implemented either alone or associated with other methods [26].

### 1.1. Thermal Processing Leads to Crystallization

Once the dielectric layers are deposited, they need to be thermally processed. Annealing in air is observed to monotonically improve stoichiometry, as well as reduce optical and mechanical losses in dielectric mirrors [27]. Unfortunately, the annealing process also gives atoms the freedom to diffuse and collect into progressively more ordered regions within the amorphous matrix. Crystal growth is observed when a threshold temperature is exceeded during annealing [28] and the film’s optical and mechanical qualities are degraded. This happens because zirconia glass is a metastable state and crystals are an energetically more favorable state. It is believed that crystallization starts from “seeds”, microscopic glass defects consisting in partially ordered regions that randomly form in various concentrations and shapes during deposition.

When the seeds are below a critical size, they are unstable and even tend to dissolve in the amorphous glass due to the competition between the energetic gain in the crystallized volume and the energetic cost of its surface. Above the critical size, however, these defects grow into crystals, provided there is sufficient atom mobility in the glass during annealing [29].

### 1.2. Optical Scattering and Thermal Noise May Be Related to Crystals

Crystals have dielectric properties which are different from the amorphous phase in the layer and, if present in sufficient numbers and sizes, they can cause significant light scattering. The fluctuation–dissipation theorem relates the thermal noise arising from the coating films to the mechanical quality factors [7,30]. The mechanical losses may be described in terms of two-level systems, which can be expected to be more frequent in the strained environment at the interface between the ordered crystallized regions than in the amorphous bulk [31].

If these suppositions are correct, the suppression of crystal formation during deposition and thermal processing can be expected to reduce both thermal noise and light scattering in mirror coatings [26,32]. Here, we pursue the idea that material interfaces constitute obstacles to the growth of crystals. Therefore, the segmentation of optical dielectric layers into thinner nano-layers may suppress crystal formation. While the nano-layered segmentation of titania with silica has already been observed to mitigate the formation of titania crystals [26], we engage in a more systematic program of exploration, with the aim of identifying material pairs and optimal segmentation designs that suppress crystal formation. If successful at repressing the formation of crystals, these optimized nano-engineered dielectric materials may enable the creation of high-refraction materials with improved optical and mechanical properties. In this paper, we report on a study of zirconia nano-layered with silica characterizing crystal formation in the course of annealing cycles.

### 1.3. Nano-Segmentation with Silica as a Performance Benchmark

At first sight, silica, a low-refraction index material, might appear as an unexpected choice to engineer a high-refraction index meta-material. However, silica is a good glass former with extremely low crystal-forming tendencies and has a high melting temperature, which makes it resistant to alloying. Its measured mechanical quality factor at room temperature (>10${}^{8}$ in highly pure molten glass form [33,34]) exceeds that of all other glasses. In the deposited glass form, the quality factor of silica is less than $10^{5}$, but still greater than other materials [20]. Conversely, virtually all candidates for high-index materials (metal oxides such as titania, zirconia, tantala, etc.) typically have quality factors smaller than $10^{4}$, lower melting points, and tend to easily crystallize, some even during deposition. Because of these properties, silica provides an optimal nano-segmentation material benchmark to compare with the performance of other higher refractive index materials.

### 1.4. Key Findings and Outline

From our results, we confirm that zirconia crystal formation starts at higher annealing temperatures in thinner nano-layers. Furthermore, the formation of crystals during annealing does not happen as a step function at a fixed temperature; instead, it progresses over a ~100 °C wide transition interval until saturation occurs after using only a fraction of the available amorphous material. These observations, taken together, suggest that the number of seed crystallites statistically occurring during deposition are limited in number, the size of the crystals that they prime is limited, and that, in the absence of seeds, the amorphous material may be unable to form crystals.

This article is organized as follows: Section 2 briefly presents the methodology from sample fabrication and annealing to X-ray diffraction (XRD) observations; Section 3 presents the observations of the segmented zirconia coating samples; Section 4 provides a summary of findings, a discussion of their possible implications, and identifies directions for future research aiming to improve the sensitivity of gravitational wave detectors.

## 2. Methodology

### 2.1. Sample Fabrication by Electron-Beam Evaporation

The nano-layered coating samples were fabricated using an OptoTech OAC-75R coater in the thin-film laboratory of the “Waves Group” of the University of Sannio in Benevento, led by Prof. Innocenzo M. Pinto [35]. The coater uses electron-beam evaporation with a quartz crystal monitor to measure the deposition rate and terminate the layer deposition upon achieving the desired thickness. An ion gun (ion assist) operating in an atmosphere with 2μTorr of argon and 8μTorr of oxygen compactified the layers during deposition while ensuring the best stoichiometry of the deposited oxide. A rotating carousel switched between oxide targets in molybdenum crucibles for the production of nano-layered coatings with alternating materials. The coating sample substrates used in the present study were 25 mm in diameter, 1 mm thick, polished fused silica wafers. Each coating started (base) and ended (cap) with a 2 nm nominal protective layer of silica.

### 2.2. Annealing Sequence and XRD Characterization

Coating samples were annealed in the air, with the coated side up, in individual stainless-steel Petri dishes. Annealing was performed in a programmable kiln with temperatures increasing at a rate of 3 °C/min. Once the nominal temperature was reached, it was kept constant for 24 h, after which the temperature control program ended, allowing the samples to cool down to room temperature inside the kiln. Annealing was performed in 50 °C increments for all samples. To monitor the formation of crystals, the samples were XRD scanned after each annealing cycle. A Bruker D2 phaser with a copper tube X-ray source emitting radiation from the K-alpha shell with a wavelength $\lambda_{X}=0.15406\;\mathrm{nm}$ was used to obtain XRD data.

X-rays probed the sample to a depth of ~100 μm, while the coating film has a total thickness of ~300 nm (see Table 1). Consequently, most of the XRD signal comes from the underlying amorphous silica substrate. On the uncoated side, this appears as a ~5° wide feature centered on ~21° in 2θ, constituting a smooth “background” profile. The coating reveals itself as features above the slightly attenuated background profile at the single digit percent level in a surface commensurate to the relative thickness to the X-ray probing depth. For this reason, both the front (coated) and back (uncoated) of the substrates were scanned so that the silica background profile could be subtracted. Subtracting the background does not take into account contributions from the silica nano-layers nor effects from attenuation. However, this does not have a significant impact on the conclusions drawn from the peak amplitude and width.

Since the coated film is thin compared to the X-ray attenuation length, the amount of crystallization was taken to be proportional to the number of X-ray counts constituting a crystallization signature peak, henceforth referred to as the “peak-count”. We used the same methodology for the measurement of the amorphous components other than silica.

Films before crystallization produce a wide peak similar to the silica substrate background but centered around a characteristic value with an amplitude proportional to the amount of coating material. The presence of crystals manifests itself as narrower peaks at angles characteristic of the crystal inter-atomic distances. Thus, a sample is considered fully glassy if the amplitude of a fitted, suitably narrow, Gaussian function centered on the crystal characteristic 2θ angle is compatible with zero up to statistical errors.

## 3. Results

The coating samples fabricated for this study consist of zirconia nano-layered with silica, with their respective thicknesses listed in Table 1.

XRD scans were initially performed on each sample as deposited to check that no crystallization occurred during deposition and to establish a reference. The range extended from 2θ = 21° to 2θ = 39° in steps of 0.02° spending 2 seconds per step. Additional XRD scans were performed after each annealing step from 300 °C to 900 °C in increments of 50 °C. These scans range from 2θ = 16.5° to 2θ = 42° in steps of 0.02° spending 1 s per step. At each annealing temperature step, the uncoated back side (back-scan) of one sample was scanned to provide data for background subtraction.

As an example, the XRD scans of the 28-layer coating sample (14 layers of 14.4nm of zirconia) are shown after annealing at 300 °C on graph A of Figure 1 and after the beginning of crystallization at 400 °C on graph B. The back-scan is scaled (gray dots in Figure 1) to match each of the sample front scans in the region extending from 2θ = 21° to 2θ = 24°, and then subtracted with a coefficient to eliminate the contribution from the fused silica substrate. In the residual, before crystallization, there remains only a broad peak centered near 30° due to the amorphous zirconia (black dots in panel A of Figure 2). Once zirconia starts to crystallize, two additional narrower peaks arise at approximately 30° and 35°, respectively, growing to the expense of the broader peak (black dots in graph B of Figure 2). These peak locations correspond to those known for crystallized zirconia [36].

The fitting of a Gaussian function (solid curves in Figure 2) provides a measurement of the peak-count and width. In the fitting procedure, the positions of the peaks are forced to their established values (from a strongly crystallized sample) to reduce the number of degrees of freedom. In order to compensate for possible instrumental throughput drifts, each peak-count is normalized to the number of X-ray counts in the region also extending from 2θ = 21° to 2θ = 24°, where zirconia does not contribute any feature in addition to the silica substrate signal. The normalized X-ray peak-count is then regarded as a quantifier for the amount of material that became crystallized.

The same analysis was carried out for the broad peak due to amorphous zirconia. In this case, both the position and the width of the peak were determined from a fit of the scan of an “as-deposited” fully glassy sample. The position and width are then imposed on all other fits leaving the peak-count as the only free fit parameter. The normalized X-ray peak-count of the wide peak was then regarded as a quantifier for the amount of material that became crystallized.

Figure 3 shows the full evolution of the 30° and 35° crystallization normalized peak-counts compared to the amorphous material peak count (25–35°) as a function of the annealing temperature for the 28-layer sample, taken as an example. We observed that the amount of crystallization increases over a range of 100 °C and plateaus past 450 °C. At the same time, the amount of amorphous material (25–35°) decreases, and plateaus at ∼50% of its initial X-ray peak-count over the same temperature range. This indicates the transfer of material from the amorphous phase to the crystalline phase. Crystallization is thus not complete, and part of the zirconia remains in the amorphous phase.

Essentially, all samples display similar trends, but with a marked shift in the crystallization threshold towards higher temperatures for thinner nano-layers of zirconia, as illustrated in Figure 4.

It can be noticed that the height of the crystallization plateaus observed in Figure 4 tends to increase with the decreasing zirconia nano-layer thickness from 26.9 nm to 3.6 nm while the accumulated thickness of zirconia decreases from 226.8 nm to 188.3 nm (cfr. Table 1). This suggests that while the segmentation frustrates crystallization, it also increases the number of seeds from which crystallization grows during annealing at a sufficiently high temperature and crystallization only extends for a limited lateral distance from the seeds.

Something different happens in samples with the thinnest zirconia nano-layers (less than 3 nm) which all start to crystallize at 800 °C and for which a plateau is not observed. We will come back to this when discussing the crystal sizes.

We define the crystallization temperature for each sample as the temperature at which the X-ray-normalized crystallization peak-count passes the 50% level of their plateau value. The plateau level is not well defined, so it is estimated to be the height of the last data point in the plateau. The crystallization temperature is shown as a function of nano-layer thickness in Figure 5; it ranges from 300 °C in the 26.9 nm thick zirconia nano-layers to over 800 °C for less than 1.9 nm-thick zirconia nano-layers. This clearly demonstrates the quenching of crystallization resulting from the segmentation.

We can clearly see the effect of segmentation on crystallization from the size of the crystals obtained by applying the Scherrer relation, with a shape factor of unity, that links the vertical size of crystal grains to the width of their diffraction peaks [37]. The Scherrer relation assumes that there is no strain on the crystal. The effect of this strain is very small relative to the observed variations, so this is a reasonable approximation. Figure 6 shows the crystal sizes as a function of the annealing temperature for the different zirconia nano-layer thicknesses. The crystal size rapidly increases with temperature until it reaches a plateau value that follows the nano-layer thickness for several samples (8.1 nm, 4.9 nm, and 3.6 nm). For the two samples with the thickest zirconia nano-layers (26.9 nm and 14.4 nm), the crystal size saturates well below the nano-layer thickness [38]. This indicates some mechanism that quenches the crystal growth in the bulk [39].

Above 800 °C, in the 4.9 nm and 3.6 nm samples, the crystal size exceeds the zirconia nano-layer thickness, indicating a breaking-down of the ∼1.7 nm silica layers and some lateral segregation [40]. The three samples with the thinnest zirconia nano-layers (1.3 nm, 1.5 nm, and 1.9 nm) directly crystallize to sizes exceeding the respective nano-layer thicknesses. The observed punch through effect indicates that for ⪅2 nm-thick silica nano-layers in zirconia, sufficient mobility for crystallization to expand through multiple layers happens at approximately 800 °C. This suggests that the crystallization would start at even higher temperatures with a thicker and more stable silica nano-layer. This indicates the necessity of a dedicated study to establish the thickness of silica nano-layers required to block against crystallization as a function of temperature.

## 4. Summary and Discussion

Most importantly, we observed that the zirconia crystallization temperature increases from ∼370 °C to above 800 °C with increasing segmentation, indicating that, as expected, nano-layering suppresses crystal formation, also in zirconia segmented by silica. This is indicative that, if thermal noise and optical scattering noise are related to the presence of crystals in mirror coatings, nano-layering could be an effective means of noise reduction.

We noticed that crystallization saturates while a significant fraction of the amorphous material remains. At the same time, and as expected, up to 800 °C, the crystal size measured along a direction perpendicular to the nano-layer remains inferior to the nano-layer thickness. However, the fact crystallization saturates suggests a mechanism is at play to quench the crystal growth from seeds within the nano-layer. It is also noted that, in the thicker nano-layers, the crystal sizes do not even reach the nano-layer thickness. This crystal growth quenching can be described as a function of the crystal size in the bulk in terms of a fourth power term in the free energy driving the crystallization [39]. The saturation of both the amount of crystallization and crystal size together suggest that the amorphous phase is meta-stable in the absence of “crystallization seeds”, the crystal growth is also quenched within the thickness, and that there is a limited number of these seeds. The fact that the relative plateauing amounts of crystallized material increase in thinner nano-layers suggests that some of the crystallization seeds are associated with the nano-layers’ interfaces.

Above = 800 °C, the crystal sizes exceed the nano-layer thickness, indicating that the 1.9 nm silica nano-layers fail to confine crystallization. It is worth noting that the ∼1700 °C melting point of fused silica would impede diffusion at higher temperatures. The high surface-to-volume ratio of thin layers reduces the melting temperature and the starting of diffusion. This suggests we should investigate the effectiveness of silica layers at blocking crystallization growth as a function of their thickness. Here, silica is used as a separator while high refraction indices are desired. Additional studies are needed to establish to which extent an alternation of high refraction materials results in a similar resistance to crystallization.

We also observed that the transition from amorphous material to crystals as a function of the annealing temperature is progressive, taking place over a ∼100 °C wide interval. It is likely that the same is true in the titania-doped tantala coatings used in LIGO. The typical procedure used to choose the annealing sequence in dielectric mirrors is to increase the annealing temperature on witness samples to find the temperature of the onset of crystallization with XRD scans. Then, the actual mirror is annealed at a temperature of ∼100 °C, which is lower together with more witness samples to check that no crystallization occurred. XRD can be used to detect only relatively large fractions of crystallized materials. Therefore, our observation suggests that even if the mirrors are annealed at a temperature 100 °C below the crystallization threshold, a limited number of crystals may be present. Those crystals could constitute the observed optical scatterers. It was previously found that a density of 100–1000 femto-light scatterers per layer, per cm${}^{2}$ in LIGO Hanford mirrors is present [9,41]. Assuming that optical scatterers are crystals similar to those observed in this work, they are in such a low concentration that XRD scanners would be completely insensitive to their presence. They collectively scatter at large angles of 10–50 ppm of the reflected light. Yamamoto reports [42] that perhaps ten times more scatterers are present in the Caltech 40 m interferometer mirrors, and other mirrors have different scatterer densities. In both cases, this is a much lower number than the crystals that form at saturated crystallization ($∼10^{10}$/cm${}^{2}$ based on the fraction of crystallized material and crystal sizes measured in the present study).

The amount of light a crystal scatters in a glassy matrix is difficult to calculate [43]. The refraction index of small crystals is not well defined due to the large surface-to-volume ratio and the fact that resonant conditions may apply. The ∼10nm crystal size falls close to the first peak of the Mie–Rayleigh theory [44,45]. However, for scatterers much smaller than a wavelength, the scattered power falls with the Rayleigh tail. The sixth power in the Mie–Rayleigh formula enters twice in the phase noise generated by scattering in the interferometer [11]. Nano-layering could reduce both the number of scatterers and more importantly the size and their individual contribution to noise.

This exploration of nano-layered materials for optical coatings needs to be complemented with correlated measurements of the coatings’ mechanical quality factor as a function of segmentation and formulation. In order to test the effectiveness for thermal noise reduction, the XRD monitoring of crystallization will test the effectiveness of nano-layered designs for scattering noise reduction, and the XRD measurements need to be complemented and correlated with optical scattering measurements akin to those used to detect the scatterers on the LIGO mirrors, illuminating and directly measuring the optical scattering in microscope inspections with techniques pioneered in the California State LA labs [41].

## Figures and Tables

**Figure 1 nanomaterials-11-03444-f001:**
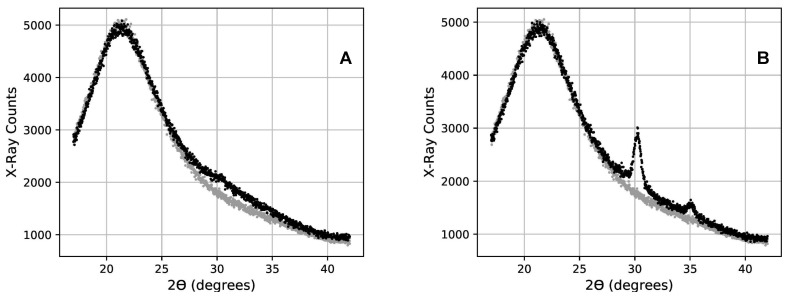
The scan of a sample’s uncoated backside (gray points) is scaled and compared to the scan of a sample’s coated front side (black points). On graph (**A**), crystallization has not started, and the noticeable broad feature is attributed to the bond length and amorphous nature of the material in the coating film. Graph (**B**) reveals crystallization in the same coating sample. This crystallization is highlighted by peaks centered at values governed by the crystal lattice constants and Bragg’s law. The amorphous component has diminished but not disappeared.

**Figure 2 nanomaterials-11-03444-f002:**
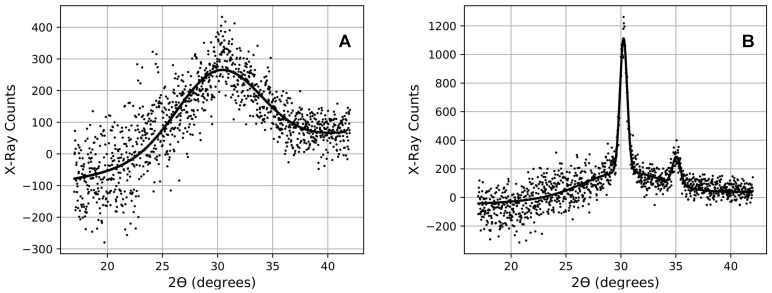
For each X-ray Diffraction (XRD) scan, the coating film contribution was obtained as the subtraction of the uncoated backside from the profile. On graph (**A**), only the amorphous component was visible while on graph (**B**), it reduced the surface to the benefit of narrower peaks, thus revealing the crystallization. An overall slope, due to the deposited coating slowly changing the X-ray attenuation, was accounted for in the fit with a first-degree polynomial in addition to the three Gaussian functions describing the individual peaks.

**Figure 3 nanomaterials-11-03444-f003:**
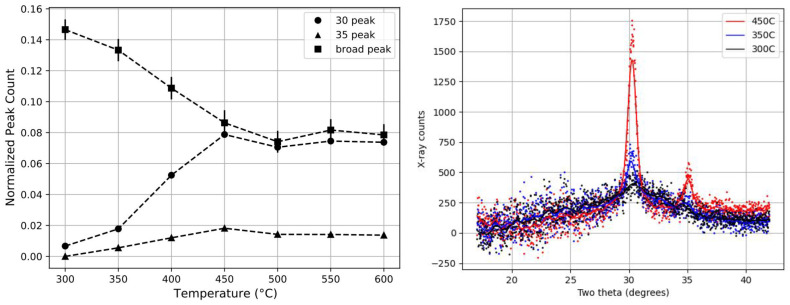
The plot on the left shows that the normalized peak-counts (defined in the text) for the crystallization and amorphous components are shown as a function of the annealing temperature for the sample with 14 layers × 14.4nm of zirconia. This demonstrates the increase in crystallization at the expense of the amorphous material without reaching the full crystallization of the available zirconia. The plot on the right demonstrates the evolution of the XRD data after annealing.

**Figure 4 nanomaterials-11-03444-f004:**
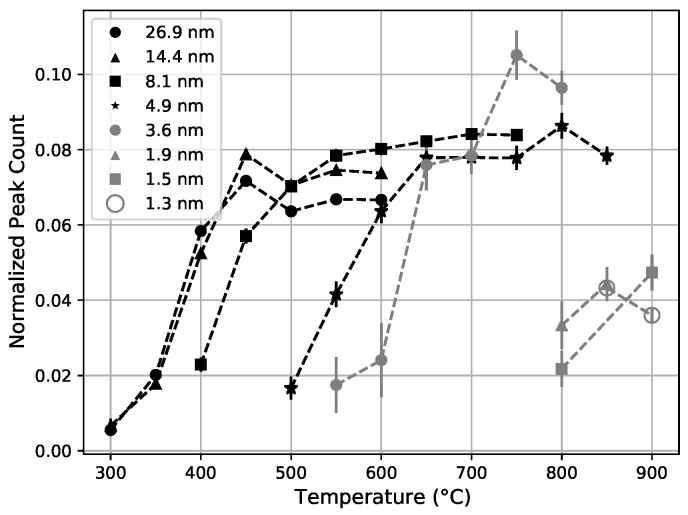
The increase in the amount of crystallization is shown as a function of the annealing temperature. A plateau is reached ∼100 °C past the onset of the crystallization. Note that the lowest temperature data point for the 1.3 nm sample overlaps with the data point for the 1.9 nm sample. The thicknesses in the legends refer to the zirconia layer thickness of the sample, described in Table 1. Points not shown for lower temperature are all consistent with zero crystallization.

**Figure 5 nanomaterials-11-03444-f005:**
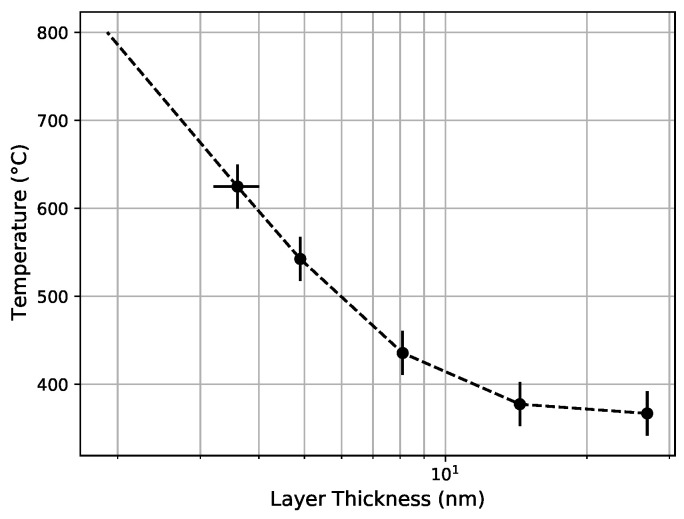
The crystallization temperature is shown as a function of the zirconia nano-layer thickness. Signs of crystallization appear at 800 °C for the thinnest layers (1.9 nm, 1.5 nm, and 1.3 nm). However, they do not fully crystallize as we explained in this paper. Because of this, they are omitted from this plot.

**Figure 6 nanomaterials-11-03444-f006:**
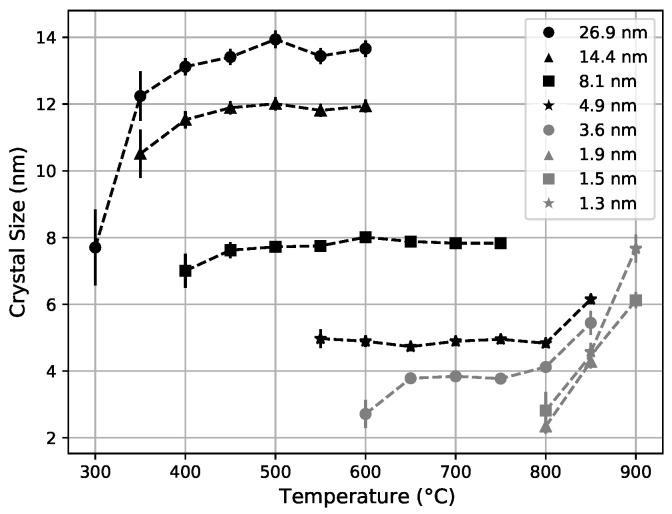
Zirconia crystal size as a function of annealing temperature. The crystal size is calculated from the crystallization peak width, with error bars representing statistical errors in the peak fits. Here, we can clearly see the breakdown of the silica barrier at 800 °C as the crystal size is larger than the zirconia layer size. The thicknesses in the legends refer to the zirconia layer thickness of the sample, described in Table 1. Points not shown for lower temperature are all consistent with zero crystallization.

**Table 1 nanomaterials-11-03444-t001:** Increasing numbers of thinner zirconia layers interleaved with fixed thickness silica designed to determine the crystallization temperature threshold for zirconia. The layer thicknesses for both zirconia and silica are average individual thicknesses.

		SiO_2_				ZrO_2_		
**Layer Count**	**Overall Thickness (nm)**	**Number of Layers**	**Layer Thickness (nm)**	**Sum of Layers (nm)**		**Number of Layers**	**Layer Thickness (nm)**	**Sum of Layers (nm)**
14	200.2	7	1.7±0.2	11.9		7	26.9±0.1	118.3
28	226.8	14	1.8±0.1	25.2		14	14.4±0.2	201.6
50	247.5	25	1.8±0.1	45.0		25	8.1±0.1	202.5
84	277.2	42	1.7±0.1	71.4		42	4.9±0.1	205.8
126	346.5	63	1.9±0.2	119.7		63	3.6±0.4	226.8
168	319.2	84	1.9±0.1	159.6		84	1.9±0.2	159.6
200	340.0	100	1.9±0.1	190.0		100	1.5±0.2	150.0
224	358.4	112	1.9±0.1	212.8		112	1.3±0.2	145.6

## Data Availability

The data underlying the results presented in this paper are not publicly available at this time but may be obtained from the authors upon reasonable request.

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
