# Peer review of "Crystallization in Zirconia Film Nano-Layered with Silica"

_nanomaterials, 2021, doi:10.3390/nano11123444_

Round 1

Reviewer 1 Report

The article entitled "Crystallization in Zirconia film nano-layered with Silica" reports on the XRD analysis of mirror coatings made of ZrO2 and SiO2 which were subject to annealing. The goal is to determine the temperature threshold above which ZrO2 crystals, which are acting as light scatterers, are growing. The topic is actually quite important for the community building and using resonant optical cavity interferometers. The paper is well written. Below are some comments and remarks:

Comments:

Line 177: "In the fitting procedure, the positions of the peaks are forced to their established values". Fitting peaks position would allow the refinement of the lattice constant and would certainly improve the peak fit.

Figure 2: -Along the same lines as the previous comment, it is unclear if the fitting in the panel B was done with three peaks. Were three peaks and a linear baseline used ? This should be shown. -Panel A and B, the difference between calculated and measured pattern needs to be included. -A Negative X-Ray count means that the intensity measured on the uncoated side is higher than on the coated side. This is not expected and is a limitation of the methodology. Author should comment on it in the experimental section.

Figure 3: -inset: "30 peak" and "35 peak" should be explained in the figure caption. -a second panel should be added to show the evolution of the XRD patterns as a function of temperature for a selected sample that is representative of all samples. -How is define the temperature threshold? Is the intersection between two linear fits (below and above the threshold) used ? "by eyes" as mentioned in the text can't be accepted.

Figure 4: For films thicker than 3.6nm, a plateau can be observed in figure 3. However for films thinner than 3.6nm, a plateau can hardly be observed. As a consequence, the corresponding Fig 4 points are not relevant and should be removed from the plot.

Line 218 and Figure 6: The Scherrer equation is used to determine the crystal size. The authors assume the absence of strain. The use of Williamson-Hall plot would be more appropriate to estimate both size and strain.

Figure 6 and Line 223-225: It is said that the crystal size increases suddenly above 500°C (for 4.9nm). Why are no crystals seen below 500°C ?

Figure 4 and 6: For 4.9nm film crystal size remains constant (between 500°C and 650°C) while the ZrO2 peak intensity increases. This indicates that more crystals with similar size are formed. It means more scatterers in the films. Between 650°C and 800°C, both peak intensity and particle size are constant. Why is Ostwald ripening not observed ? Similar discussion should be included for all the films, so a general trend can be drawn.

In conclusion, the paper needs revision before acceptance.

Reviewer 2 Report

The manuscript “Film Crystallization of Zirconia Nanolayers with Silica is an interesting work. In this, the crystalline growth of zirconia crystals in nanolayers alternated with silica is analyzed as a function of the thickness of the nanolayer and the annealing temperature. The work is well structured and the conclusions are supported by the results.
In order to improve understanding, a clarifying scheme of the nano-layer structure is lacking.
In my opinion, the manuscript can be accepted for publication in Nanomaterials.

Author Response

Thank you for taking the time to review our paper and for your support in publication. We have made some changes to the paper based on comments from other reviewers. These changes are marked in red on the manuscript.

Reviewer 3 Report

The authors reported a behavior of crystallization reduction in ZrO2 by SiO2 nanolayering. The work is beneficial for the gravitational wave study and could have the potential to improve the performance of instruments accordingly.  XRD is the main characterization tech for analyzing the material. I will recommend the manuscript for publication after the questions below have been addressed.

(1) For the abstract, line 5 crystal should be in lower case. I personally feel the abstract is not that complete after line 7-8 “However” sentence, compared with the detailed summary and discussion. Maybe need one more sentence for the summary.

(2) In the introduction, two major noise ways use the numbering 1,2 which might confuse with introduction 1, and subsection 1.1,1.2, I suggest adjusting the numbering system but it is not required.

(3) For the methodology and results sections, it looks like XRD is the main testing method e-beam coating and annealing process. How does other research work characterize similar coatings? Is the XRD the only method and have you used other methods to analyze the nano-layered coating samples? There are multiple methods for analyzing the films’ components, morphology, etc…  

(4) For Table 1, how do you define the layer thickness, and do you have proof of why is in that range? Why do you choose these specific layer counts 14,28, 50,… Similarly, how to achieve the thickness data in Figures 4,5,6?

Round 2

Reviewer 1 Report

The authors have made the expected changes to the manuscript and the answers to my questions/comments are satisfactory.

I recommend the publication of the manuscript